# De Ritis Ratio to Predict Clinical Outcomes of Intermediate- and High-Risk Pulmonary Embolisms

**DOI:** 10.3390/jcm13072104

**Published:** 2024-04-04

**Authors:** Koray Durak, Katharina Nubbemeyer, Rashad Zayat, Jan Spillner, Slavena Dineva, Sebastian Kalverkamp, Alexander Kersten

**Affiliations:** 1Department of Thoracic Surgery, Faculty of Medicine, RWTH University Hospital, 52074 Aachen, Germany; knubbemeyer@ukaachen.de (K.N.); r.zayat@bbtgruppe.de (R.Z.); jspillner@ukaachen.de (J.S.); slavena.dineva@luks.ch (S.D.); skalverkamp@ukaachen.de (S.K.); 2Department of Cardiology, Pneumology, Angiology, and Intensive Care, Faculty of Medicine, RWTH University Hospital, 52074 Aachen, Germany; akersten@ukaachen.de

**Keywords:** pulmonary embolism, transaminases, liver function, intensive care medicine, heart failure, shock, lung emboly, De Ritis

## Abstract

**Background**: Abnormal liver function tests can identify severe cardiopulmonary failure. The aspartate transaminase/alanine transaminase (AST/ALT) ratio, or the De Ritis ratio, is commonly used to evaluate acute liver damage. However, its prognostic value in pulmonary embolism (PE) is unknown. **Methods**: Two cohorts, including patients with intermediate- and high-risk PEs, were established: one with an abnormal baseline AST/ALT ratio (>1) and another with a normal baseline AST/ALT ratio (<1). The primary outcome was a 60-day mortality. Secondary outcomes included peak N-terminal pro-brain-natriuretic-peptide (NT-proBNP) levels, complications, and the need for critical care treatment. To assess the effect of abnormal AST/ALT ratios, inverse probability weighted (IPW) analyses were performed. **Results**: In total, 230 patients were included in the analysis, and 52 (23%) had an abnormal AST/ALT ratio. After the IPW correction, patients with an abnormal AST/ALT ratio had a significantly higher mortality rate and peak NT-proBNP levels. The relative risks of 60-day mortality, shock development, use of inotropes/vasopressors, mechanical ventilation, and extracorporeal life support were 9.2 (95% confidence interval: 3.3–25.3), 10.1 (4.3–24), 2.7 (1.4–5.2), 2.3 (1.4–3.7), and 5.7 (1.4–23.1), respectively. **Conclusions**: The baseline AST/ALT ratio can be a predictor of shock, multiorgan failure, and mortality in patients with a pulmonary embolism.

## 1. Introduction

Pulmonary embolism (PE) is associated with high mortality rates due to the development of obstructive shock with right heart failure and, respiratory failure and other complications [1]. Multiple prognostic models for classifying the mortality risk based on clinical parameters, such as the simplified Pulmonary Embolism Severity Index (sPESI) and PESI, are available [2]. However, hemodynamic instability can occur delayed, and a high proportion of intermediate-risk patients develop severe complications after the first day of PE diagnosis [3]. Therefore, risk stratification and decision-making for thrombolytic therapies or intensive care in this patient group remain challenging.

Elevated transaminase levels are frequently observed in patients with acute heart failure, and they are associated with clinical signs of hypoperfusion and unfavourable short-term prognosis [4,5]. This phenomenon can be explained by two different mechanisms, which are as follows: right/backward heart failure, leading to liver congestion, and left/forward heart failure, resulting in hypoperfusion of the liver, hypoxemia, and necrosis [6]. The Society for Cardiovascular Angiography and Interventions (SCAI) classified the stages of cardiogenic shock from stage A to E according to the severity and mortality risk [7]. Machine-learning algorithms revealed three distinct clusters in cardiogenic shock (non-congested [I], cardiorenal [II], and cardiometabolic [III]) [8]. Interestingly, the cardiometabolic shock cluster is characterised by abnormalities in the liver parameters and is most significantly associated with a high risk of stage D or E shock and in-hospital mortality, regardless of the cause.

The aspartate transaminase/alanine transaminase (AST/ALT) ratio, or the De Ritis ratio, has been a useful diagnostic and prognostic tool for liver diseases for several years [9]. Liver cells contain both AST and ALT in the cytoplasm. On the other hand, mitochondria within liver cells contain only AST. Slight damage to liver cells causes only a rupture of the cell wall and an equal release of AST and ALT. Stronger damage and necrosis are characterised by an additional eruption of mitochondria and cause higher levels of AST than ALT in the bloodstream. Therefore, the ratio yields information about the time course and burden of diseases by comparing values from the relatively short half-life of AST (18 h) to the longer half-life of ALT (36 h).

Our recently published case report on fulminant septic cardiomyopathy showed that an acute increase in liver transaminase levels with an abnormal AST/ALT ratio were the initial signs of cardiopulmonary failure. Further, this report revealed that intensive therapies, such as inotropes, vasopressors, or ECLS, are required [10]. However, previous studies on the diagnostic value of transaminase levels for detecting early-stage hemodynamic deterioration are limited.

We hypothesised that the baseline AST/ALT ratio is a simple tool for detecting hemodynamic deterioration in patients with intermediate- and high-risk PEs and can be a predictor of clinical outcomes and the need for critical care treatment. The current study aimed to investigate the effect of an abnormal AST/ALT ratio on the clinical course of a PE.

## 2. Materials and Methods

### 2.1. Inclusion and Exclusion Criteria

This retrospective, single-centre study used data from the hospital records and the International Classification of Diseases, Tenth Revision (ICD-10-CM) codes *415.1x. All patients aged ≥18 years who presented with intermediate- and high-risk PEs (ICD-code I26.0) and who were admitted to a tertiary healthcare centre between January 2007 and January 2022 were included in the analysis. All cases identified automatically were confirmed to have a diagnosis of PE based on a computed tomography pulmonary angiogram, which was available for review in our electronic medical record. Patients who did not undergo diagnostic imaging based on the data from our system, such as those transferred from another facility who did not have data uploaded in our system, and patients without data on the AST and ALT levels on the day of a PE diagnosis, were excluded from the study.

Two cohorts were retrospectively established: one with a normal AST/ALT ratio and another with an abnormal AST/ALT ratio. An abnormal AST/ALT ratio was defined as elevated transaminase levels (≥50 U/L in men and ≥35 U/L in female patients) with an AST/ALT ratio of ≥ 1 based on the traditional cut-off value for alcoholic liver disease [11].

### 2.2. Data Collection

We extracted data on the demographic characteristics of the patients (i.e., age, sex, and medical history, including cardiovascular and pulmonary diseases, comedication, and laboratory parameters), complications at baseline (shock, respiratory insufficiency, and deep venous thrombosis), PE therapies, including ultrasound-assisted thrombolysis and mechanical circulatory support, and patient outcomes (in-hospital mortality, cardiac and non-cardiac organ dysfunction, infarction pneumonia, peak-brain natriuretic peptide (BNP) levels, and the need for critical care therapies). We identified all patients who received positive inotropic or vasopressor agents, which included sympathomimetics (dopamine, norepinephrine, noradrenaline, and dobutamine), PDE3 inhibitors (milrinone and amrinone), vasopressin, and calcium sensitizers (levosimendan).

Right heart systolic dysfunction (pressure and/or volume overload) was assessed by computer tomography and echocardiography. We used the following parameters and cut-off values: a right ventricular dilation or basal diameter (RVEDD) of >44 mm, a tricuspid annular plane systolic excursion of <17 mm, and ventricular interdependence according to a septal shift or D-shaped left ventricle. The use of an age-adjusted high sensitivity troponin T cut-off value (≥14 pg/mL^−1^ for patients aged <75 years and ≥45 pg/mL^−1^ for patients aged ≥75 years) was used to determine patients with elevated troponin.

Both right heart systolic dysfunction and troponin T were used to stratify the patients into intermediate low-, intermediate high-, and high-risk, according to the classification of pulmonary embolism severity and risk of early death in Table 9 of the latest guidelines [2]. Therefore, all patients with hemodynamic instability or shock at admission were classified as high risk. Patients with signs of right heart systolic dysfunction (on computer tomography or echocardiography) and troponin elevation were classified as intermediate high-risk. Patients with only one or no indicator were classified as intermediate low-risk.

Details on the PE diagnosis, transaminase levels on the day of diagnosis, mortality, and development of shock (with exact date) were extracted individually for each case and tested for accuracy using the physicians’ notes and computer tomography reports by radiologists. The trial was approved by the local ethics committee (379/19). Due to the retrospective nature of the study, the need for informed consent was waived. Patients or the public were not involved in the design, conduct, reporting, or dissemination plans of our research.

### 2.3. Primary and Secondary Outcomes

The primary outcome was a 60-day mortality after a PE diagnosis. The secondary outcomes included highest/peak N-terminal pro-brain natriuretic peptide (NT-proBNP) levels during the clinical course, development of circulatory shock, respiratory or kidney failure, a need for dialysis, invasive mechanical ventilation, vasopressor or positive inotropic or vasopressor treatment, and extracorporeal life support.

### 2.4. Statistical Analysis

Categorical variables were presented as absolute numbers and percentages. Continuous variables with normal distribution were assessed using the Kolmogorov–Smirnov test and were presented as the median and interquartile range (IQR). For a direct comparison between the two groups, the Mann–Whitney U test and Fisher’s exact test were utilised to assess the continuous and categorical variables, respectively. The percentages of missing data per variable were determined, and only two variables were affected (Appendix A). The overall percentage of missing data in all variables was <5% and the percentage of missing data in the two affected variables was not between 10–20%. Therefore, multiple imputation or other methods were not applied.

In the first comparison between patients with normal versus abnormal AST/ALT ratios, univariate analyses were performed. Then, Kaplan–Meier survival estimates with the log-rank test were used. For the second comparison between cohorts with normal and abnormal AST/ALT ratios, inverse probability weighting (IPW) was used to correct the confounding variables. These variables were identified based on the directed acyclic graph that was created prior to the data analysis (Appendix A) and the classification of pulmonary embolism severity and risk of early death [2]. To the best of our knowledge, this method is currently recommended to control confounding in causal interference studies [12]. The propensity score of an abnormal AST/ALT ratio was calculated based on the identified confounding variables. Inverse weighting was applied to this propensity score, thereby creating a pseudo-population in which the weighted averages reflect averages in the true population. This method was selected because it reduces confounding bias and is superior to the traditional propensity-matching procedures in terms of efficiency [13]. Cases with an inverse propensity score of > 15 were excluded from further analysis and would, therefore, have a too large effect compared to other cases: zero cases were deleted. To evaluate the covariate balance and to confirm whether this pseudo-population confounding was successfully removed, the standardised mean differences of confounding variables were calculated prior to and after weighting (Appendix A). To assess the primary and secondary outcomes, the causal relative risks of the weight-adjusted dataset were evaluated. All statistical comparisons were two-sided, and *p*-values of <0.05 indicated statistically significant differences. Statistical analysis was performed using R statistics in the R Studio interface (version 2022.07.2) [14].

## 3. Results

### 3.1. Demographic Characteristics of the Participants

In total, 293 patients were included in the database. During the individual confirmation process, 63 were excluded from further analysis because of an initial error in ICD-code identification (n = 62) or double identification (n = 1). Finally, 230 patients admitted to a tertiary healthcare centre due to having a PE during a 15-year period were finally included in the analysis (Figure 1).

The overall median age of the patients was 67 (interquartile range [IQR]: 55–77) years. A total of 42.1% (n = 100) of patients were women. The most common comorbidities were obesity, diabetes mellitus type 2, and chronic left heart failure (19.1%, 17.4%, and 17%, respectively). Complications observed on the day of PE diagnosis were right heart systolic dysfunction (62.2%, n = 143), deep venous thrombosis (45.2%, n = 104), and shock (13.9%, n = 32). The median length of hospital stay (LOS) was 10 (IQR: 6–16) days. Approximately 55.7% (n = 128) of the patients underwent ultrasound-assisted thrombolysis (USAT) for PE treatment. Table 1 depicts a complete overview of the demographic characteristics of the patients.

### 3.2. Normal Versus Abnormal AST/ALT Ratio before IPW

The baseline demographic characteristics of patients with normal (n = 178) and abnormal (n = 52) AST/ALT ratios were compared. Patients with abnormal AST/ALT ratios were significantly younger (age: 61.5 [IQR: 52–73] vs. 68.5 [IQR: 57–77] years, *p* = 0.023), were less likely to receive USAT for PE (32.7% vs. 62.4%, *p* < 0.001), and had a significantly higher incidence rate of shock at baseline (36.5% vs. 7.3%, *p* < 0.001) than patients with normal AST/ALT ratios. Most patients with abnormal AST/ALT ratios consisted of the high-risk group (36.5%, n = 19), whereas patients with normal AST/ALT ratios mostly belonged to the intermediate low-risk group (50.6%, n = 90). Furthermore, chronic left heart failure and coronary artery disease were significantly more frequent in patients with abnormal AST/ALT ratios (Table 1).

As shown in Table 2, patients with an abnormal AST/ALT ratio had a significantly higher 60-day mortality rate than those with a normal AST/ALT ratio (38.5% vs. 2.8%, *p* < 0.001). This finding was confirmed using the Kaplan–Meier survival estimates, as depicted in Figure 2 (*p* < 0.001 using the log-rank test). Regarding the secondary outcomes, patients with an abnormal AST/ALT ratio had significantly higher incidence rates of shock (32.7% vs. 3.9%, *p* < 0.001), acute bleeding-related anaemia (42.3% vs. 18%, *p* = 0.001), acute kidney failure (40.4% vs. 16.3%, *p* < 0.001), and cardiac arrest (28.8% vs. 7.9%, *p* < 0.001) than those with a normal AST/ALT ratio. Patients with an abnormal AST/ALT ratio had significantly higher peak-BNP levels (3700 vs. 2550 pg/mL, *p* = 0.023) than those with a normal AST/ALT ratio. Furthermore, patients with an abnormal AST/ALT ratio were significantly more likely to receive inotropes or vasopressors (40.4% vs. 11.2%, *p* < 0.001), mechanical ventilation (61.5% vs. 16.9%, *p* < 0.001), dialysis (19.2% vs. 3.9%, *p* = 0.001), and ECLS (13.5% vs. 1.7%, *p* = 0.001) than those with a normal AST/ALT ratio.

### 3.3. Normal Versus Abnormal AST/ALT Ratio after IPW

The covariate balance improved for risk classification and nearly all confounders after IPW (Appendix A). There were no significant differences in all the baseline characteristics of patients in the pseudo-population after IPW, as shown in Table 3. Patients with an abnormal AST/ALT ratio had a significantly higher 60-day mortality rate than those with a normal AST/ALT ratio (25.7% vs. 2.8%, *p* < 0.001), as shown in Table 4. Regarding the secondary outcomes, patients with an abnormal AST/ALT ratio had significantly higher incidence rates of shock (37.8% vs. 3.7%, *p* < 0.001), acute bleeding-related anaemia (39.1% vs. 21.6%, *p* = 0.044), and acute kidney failure (40.4% vs. 20.8%, *p* = 0.027) than those with a normal AST/ALT ratio. Patients with an abnormal AST/ALT ratio had significantly higher peak-BNP levels than those with a normal AST/ALT ratio (3909 vs. 2400 pg/mL, *p* = 0.021). Furthermore, patients with an abnormal AST/ALT ratio were significantly more likely to receive inotropes and vasopressors (30.3% vs. 11.1%, *p* = 0.003), dialysis (21% vs. 7.2%, *p* = 0.048), mechanical ventilation (52.2% vs. 22.8%, *p* = 0.002), and extracorporeal life support (9.7% vs. 1.7%, *p* = 0.006) than those with a normal AST/ALT ratio.

### 3.4. Causal Relative Risk of Primary and Secondary Outcomes after IPW

After IPW, in patients with abnormal AST/ALT ratios, the causal relative risk of 60-day mortality, shock, acute kidney failure, treatment with inotropes and/or vasopressors, mechanical ventilation, and support-veno-arterial extracorporeal membrane oxygenation was significantly increased at 9.2 (95% confidence interval: 3.3–25.3), 10.1 (4.3–24), 1.9 (1.1–3.4), 2.7 (1.4–5.2), 2.3 (1.4–5.2), 5.7 (1.4–23.1), respectively (Figure 3). The relative risk for being treated with continuous renal replacement therapy was not significantly increased with 2.9 (0.97-8.6).

## 4. Discussion

### 4.1. Main Findings

This single-centre study evaluated the effect of abnormal AST/ALT ratios at baseline on various clinical outcomes in 230 patients with intermediate- and high-risk PEs. The following results were obtained: First, patients with an abnormal AST/ALT ratio had a significantly higher rate of mortality, shock, multiorgan failure, and intensive care therapies than those with a normal AST/ALT ratio. Second, patients with an abnormal AST/ALT ratio had significantly higher peak-BNP levels during the clinical course than those with a normal AST/ALT ratio. Third, the primary outcome and most secondary outcomes remained significant after the IPW correction.

### 4.2. Previous Studies on the Role of Transaminase Levels in PE

The number of studies on the influence of transaminases on PE outcomes is limited. To the best of our knowledge, our study investigated the largest cohort regarding this specific hypothesis. Aksoy et al. investigated the AST/ALT ratio of patients with an acute PE. The results showed that an AST/ALT ratio of >1.3 was associated with mortality, and its prognostic sensitivity and specificity were 61% and 65%, respectively [15]. Aslan et al. found that patients with high-risk PEs were more likely to present with elevated transaminase levels than patients with intermediate-risk PEs. Hence, a high transaminase level was associated with a lower partial pressure of oxygen (PO2) and oxygen saturation [16]. However, the sample sizes of these studies were small, and clinically relevant complications or the need for intensive care therapies were not investigated.

### 4.3. Risk Stratification, Echocardiography, and Biomarkers in Patients with PEs

The PESI score, which is commonly used, can help identify a PE with a certain increased risk. Hence, low-risk patients might be treated outside the hospital, and the appropriate therapies for intermediate- or high-risk patients should be evaluated [2].

Some additional parameters are well investigated and can be added to the PESI score or other prognostic models [17,18]. The addition of echocardiographic assessment findings to common clinical parameters can improve the outcome prediction in acute PEs [19]. Grifoni et al. reported that the incidence of PE-related shock and the in-hospital mortality in patients with PEs and right heart systolic dysfunction on echocardiography is 10% and 5%, respectively [20]. However, the echocardiographic parameters can vary over the disease course and can present with intra- or interobserver variability, particularly in cases with poor visualisation [21]. Furthermore, echocardiography needs the consultation of experienced staff.

Biomarkers can be assessed using a cost-effective, simple, and routinely used method. Therefore, they might be offering a useful addition in risk stratification for predicting right heart failure and hemodynamic deterioration. Moreover, some retrospective studies investigated troponin and BNP levels, which can help identify patients at high risk for right heart failure and mortality [2,22]. The AST/ALT ratio could offer a new or rediscovered biomarker, which might be useful in identifying progressive hemodynamic deterioration.

### 4.4. Bleeding and Haemorrhagic Shock Complications

Patients with an abnormal AST/ALT ratio had significantly higher incidences of anaemia due to acute bleeding and haemorrhagic shock. Patients with PEs were frequently treated with unfractionated heparin with an activated partial thromboplastin time between 50–70 s or an adequate dosage of low-molecular-weight heparin (LMWH). However, the possible explanations for the higher incidence of bleeding are as follows: Acute liver injury leads to an early and substantial reduction in clotting factors, particularly factors VII and V. This can lead to the rapid development of coagulopathy, specifically due to the short half-life of these coagulation factors [23]. Both chronic and acute liver failure are associated with a higher risk of bleeding and are commonly an issue in invasive procedures [24,25]. The combination of an abnormal AST/ALT ratio and the use of thrombolysis in patients with PEs may cause coagulopathy and may be associated with a higher risk of bleeding.

### 4.5. Implications

Further investigation of the AST/ALT ratio as a laboratory biomarker to provide prognostic information on PE outcomes is warranted. Our results should be re-evaluated in prospective cohort studies. Eventually, the use of transaminases to guide treatment decisions in randomised controlled trials should be investigated to formulate recommendations for clinician decision-making.

Selection of the appropriate treatment for patients with intermediate- to high-risk PEs remains difficult and without clear criteria for eligibility in a high proportion of cases. Although systemic thrombolysis is used in most cases, treatment options include systemic thrombolytic therapy, catheter-directed mechanical aspiration or USAT, and surgical thrombectomy. Most hospitals obtain the transaminase parameters routinely in every patient on the day of admission. Particularly, treatment decisions by pulmonary embolism response teams (PERTs) could be positively influenced by this biomarker. Patients with an abnormal AST/ALT ratio at baseline should be closely monitored for the development of multiorgan failure and/or shock to initiate advanced therapies with a low threshold.

Our results showed that patients with an abnormal AST/ALT ratio are at higher risk for acute bleeding-related anaemia and haemorrhagic shock. Therefore, the early use of catheter-directed USAT, rather than systemic thrombolysis, should be considered and further investigated in these patients due to the following reasons: USAT may be associated with a lower risk of major bleeding complications [26], and it is more effective in reducing right heart strain and lowering pulmonary artery pressures. Therefore, it can prevent the development of shock and multiorgan failure in patients with PE [26,27].

### 4.6. Limitations

Although our results can improve our knowledge of the management of patients with a PE, the current study had several limitations. That is, retrospective studies are associated with a potential risk of selection bias, and the results are based on accurate recordkeeping. We only reported a single-centre experience, and this might affect the generalisability of the results. On a statistical level, most standardised mean differences for the covariates improved after weighting, and they were <0.1. However, the standardised mean differences of 5 of 19 covariates had an inverse propensity score between 0.1 and 0.3 after the IPW (Appendix A). Furthermore, due to the small sample size and high number of covariates, the confidence intervals were not commonly wide (Figure 3). Only six patients had an inverse weighting score between 10 and 15. This could have a proportionally large effect, but we did not truncate or remove these weights because the increase in precision may outweigh the bias induced by it [28].

Troponin and echocardiographic or computer tomography measurements for right heart systolic dysfunction could be extracted as numerical values and were conducted routinely for all hospital admissions with a pulmonary embolism. Interobserver variability can affect our variable for right heart dysfunction.

We did not have the documentation of the pre-admission vital parameters. For this reason, it was not possible to calculate the mortality risk based on the original or simplified Pulmonary Embolism Severity Index. Reporting of hemodynamic instability and diagnosis of shock is conducted retrospectively and can be vague or differ between individuals. This variable is dependent on accurate recordkeeping, and there is a risk of selection bias and confounding.

Ultrasound-assisted thrombolysis was used as a baseline variable because it is a clinical decision that is based on various subjective and objective factors that cannot be retrospectively extracted. Therefore, we did not use USAT as an outcome value so as to avoid selection bias.

## 5. Conclusions

We found that high transaminase levels with an abnormal AST/ALT ratio (>1) can be an early sign of hemodynamic deterioration and can predict various outcomes in patients with intermediate- and high-risk PEs.

Hence, this biomarker should be considered in pulmonary embolism response teams (PERTs), and patients should be closely monitored to initiate timely critical care treatment to prevent shock and multiorgan failure.

Further investigation in prospective cohort studies and the use of transaminases to guide treatment decisions in randomised controlled trials are needed. Secondly, our results revealed a significantly higher bleeding risk in patients with an abnormal AST/ALT ratio, which can be explained by multiple pathophysiologic mechanisms. We suggest to particularly investigate the prognostic value and influence of different treatment options.

## Figures and Tables

**Figure 1 jcm-13-02104-f001:**
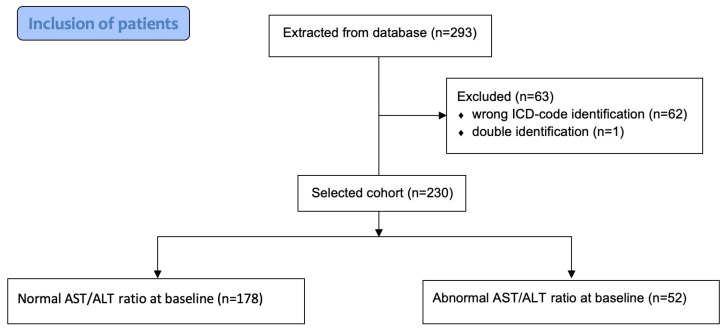
Inclusion of patients.

**Figure 2 jcm-13-02104-f002:**
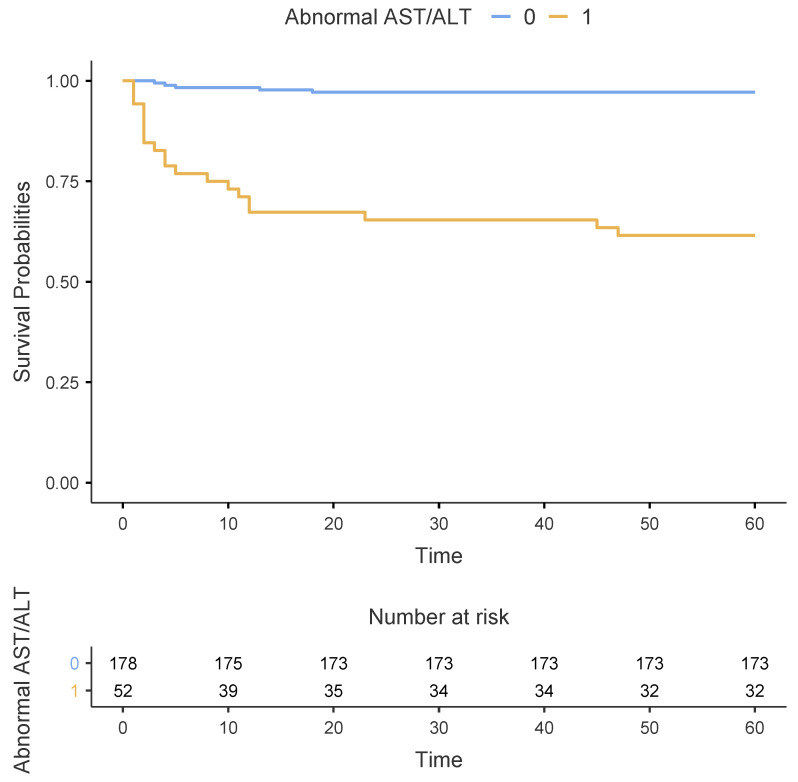
Kaplan–Meier estimates of survival between patients with normal (n = 178) and abnormal (n = 52) AST/ALT ratio during the first 60 days after PE diagnosis.

**Figure 3 jcm-13-02104-f003:**
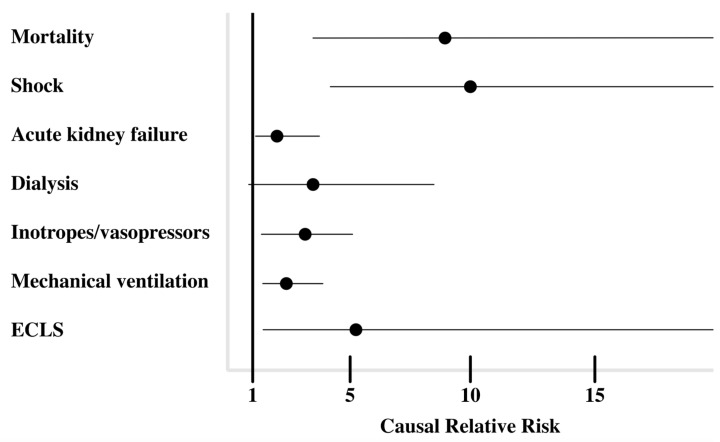
Causal relative risk of clinical outcomes after inverse probability weighting.

**Table 1 jcm-13-02104-t001:** Baseline characteristics of the participants.

	Total Number of Participants (n = 230)	Patients with Normal AST/ALT Ratio (n = 178)	Patients with Abnormal AST/ALT Ratio (n = 52)	*p*-Value
Age (years)	67 (55–77)	68.5 (57–77)	61.5 (52–73)	0.023 *
Female sex, n (%)	97 (42.1)	75 (42.1)	22 (42.3)	1.000
Total LOS (days)	10 (6–16)	10 (6–15)	11.5 (5–25)	0.234
Coronary artery disease, n (%)	9 (3.9)	4 (2.2)	5 (9.6)	0.045 *
Prior CABG surgery, n (%)	5 (2.2)	3 (1.7)	2 (3.8)	0.690
Arterial hypertension, n (%)	33 (14.3)	24 (13.5)	9 (17.3)	0.640
COPD, n (%)	1 (0.4)	2 (1.1)	0 (0)	0.834
Diabetes mellitus type 2, n (%)	40 (17.4)	34 (19.1)	6 (11.5)	0.290
Chronic kidney disease, n (%) ^†^	31 (13.5)	24 (13.5)	7 (13.5)	1.000
Chronic left heart failure, n (%)	39 (17)	21 (11.8)	18 (34.6)	<0.001 *
Liver cirrhosis, n (%)	0 (0)	0 (0)	0 (0)	NA
Obesity, n (%)	44 (19.1)	36 (20.2)	8 (15.4)	0.562
Anticoagulation before PE, n (%)	16 (7)	12 (6.7)	4 (7.7)	1.000
LVAD for chronic heart failure, n (%)	1 (0.4)	0 (0)	1 (1.9)	0.512
Deep venous thrombosis, n (%)	104 (45.2)	86 (48.3)	18 (34.6)	0.112
Right heart systolic dysfunction, n (%) ^§^	143 (62.2)	105 (59)	38 (73.1)	0.093
Shock, n (%) ^§§^	32 (13.9)	13 (7.3)	19 (36.5)	<0.001 *
obstructive, n (%)	32 (13.9)	13 (7.3)	19 (36.5)	<0.001 *
septic, n (%)	2 (0.9)	0 (0)	2 (3.8)	0.075
hemorrhagic, n (%)	3 (1.3)	1 (0.6)	2 (3.8)	0.254
Pleural effusion, n (%)	39 (17)	29 (16.3)	10 (19.2)	0.774
Infarction pneumonia, n (%) ^††^	28 (12.1)	22 (12.4)	6 (11.5)	1.000
Baseline troponin T (µg/L)	56 (27–138)	52 (26–117)	83.5 (33.3–241)	0.061
USAT for PE, n (%)	128 (55.7)	111 (62.4)	17 (32.7)	<0.001 *
Intermediate low-risk, n (%)	106 (46.1)	90 (50.6)	16 (30.8)	0.018 *
Intermediate high-risk, n (%)	92 (40)	75 (42.1)	17 (32.7)	0.288
High-risk, n (%)	32 (13.9)	13 (7.3)	19 (36.5)	<0.001 *

Continuous variables were presented as median and interquartile range (IQR) unless indicated as percentages. * *p*-values of <0.05 were considered significant. ^†^ Including all patients with an MDRD-GFR of <60 mL/min. ^††^ Including patients who later developed this complication. ^§^ Right heart systolic dysfunction on computer tomography or echocardiography. ^§§^ Shock on the day of PE diagnosis. Abbreviations: ALT—alanine transaminase; AST—aspartate transaminase; CABG—coronary artery bypass graft; COPD—chronic obstructive pulmonary disease; LOS—length of hospital stay; NA—not available; PE—pulmonary embolism; USAT—ultrasound-assisted thrombolysis.

**Table 2 jcm-13-02104-t002:** Patient outcomes.

	Total Number of Participants (n = 230)	Patients with Normal AST/ALT Ratio (n = 178)	Patients with Abnormal AST/ALT Ratio (n = 52)	*p*-Value
Mortality, n (%) ^†^	25 (10.9)	5 (2.8)	20 (38.5)	<0.001 *
Maximum NT-proBNP level (pg/mL)	2800 (1200–6875)	2550 (1146–5625)	3700 (1750–9891)	0.023 *
Shock, n (%)	24 (10.4)	7 (3.9)	17 (32.7)	<0.001 *
obstructive, n (%)	14 (6.1)	1 (0.6)	13 (25)	<0.001 *
septic, n (%)	3 (1.3)	2 (1.1)	1 (1.9)	1.000
haemorrhagic, n (%)	9 (3.9)	5 (2.8)	4 (7.8)	0.222
Acute bleeding-related anaemia, n (%)	54 (23.5)	32 (18)	22 (42.3)	0.001 *
Acute kidney failure, n (%) ^§^	50 (21.7)	29 (16.3)	21 (40.4)	<0.001 *
Cardiac arrest, n (%)	29 (12.6)	14 (7.9)	15 (28.8)	<0.001 *
Treatment with inotropes or vasopressors, n (%)	41 (17.8)	20 (11.2)	21 (40.4)	<0.001 *
Treatment with mechanical ventilation, n (%)	62 (27)	30 (16.9)	32 (61.5)	<0.001 *
Duration of mechanical ventilation (h)	0 (0–2)	0 (0–0)	55 (0–195)	<0.001 *
Treatment with VA-ECMO, n (%)	10 (4.3)	3 (1.7)	7 (13.5)	0.001 *
Treatment with CRRT, n (%)	17 (7.4)	7 (3.9)	10 (19.2)	0.001 *

Continuous variables were presented as median and interquartile range (IQR) unless indicated as percentages. * *p*-values of <0.05 were considered significant. ^†^ In-hospital mortality. All patients were discharged to a rehabilitation centre, general ward at another hospital, or home. ^§^ All patients presented with stage 2 or 3 acute kidney failure based on the KDIGO guidelines. Abbreviations: ALT—alanine transaminase; AST—aspartate transaminase; BNP—B-type natriuretic peptide; CRRT—continuous renal replacement therapy; VA-ECMO—veno arterial extracorporeal membrane oxygenation.

**Table 3 jcm-13-02104-t003:** Baseline characteristics after inverse probability weighting.

	Patients with Normal AST/ALT Ratio (n = 230.4)	Patients with Abnormal AST/ALT Ratio (n = 234.34)	*p*-Value
Age (years)	67 (55–77)	61 (53.3–71)	0.076
Female sex, n (%)	93.5 (40.6)	76.3 (32.6)	0.373
Total LOS (days)	10 (7–17)	12 (7.4–33.4)	0.218
Coronary artery disease, n (%)	11.7 (5.1)	11 (4.7)	0.915
Prior CABG surgery, n (%)	8.9 (3.9)	7.1 (3)	0.800
Arterial hypertension, n (%)	30.3 (13.1)	24.7 (10.5)	0.610
COPD, n (%)	2.3 (1)	0 (0)	0.159
Diabetes mellitus type 2, n (%)	38.7 (16.8)	44.1 (18.8)	0.819
Chronic kidney disease, n (%) ^†^	28.8 (12.5)	27.9 (11.9)	0.927
Chronic left heart failure, n (%)	28.1 (12.2)	55 (23.5)	0.070
Liver cirrhosis, n (%)	0 (0)	0 (0)	NA
Obesity, n (%)	43.1 (18.7)	42.1 (18)	0.925
Anticoagulation before PE, n (%)	15.6 (6.8)	14.8 (6.3)	0.916
LVAD for chronic heart failure, n (%)	0 (0)	4.1 (1.7)	0.327
Deep venous thrombosis, n (%)	107.2 (46.5)	94.9 (40.5)	0.545
Right heart systolic dysfunction, n (%) ^§^	136.4 (59.2)	148 (63.2)	0.697
Shock, n (%) ^§§^	32.5 (14.1)	31.2 (13.3)	0.876
obstructive, n (%)	32.5 (14.1)	31.2 (13.3)	0.876
septic, n (%)	0 (0)	4.7 (2)	0.180
hemorrhagic, n (%)	2.9 (1.2)	3.1 (1.3)	0.957
Pleural effusion, n (%)	38.3 (16.6)	36.6 (15.6)	0.880
Infarction pneumonia, n (%) ^††^	107.2 (46.5)	94.9 (40.5)	0.545
Baseline troponin T (µg/l)	53 (26-122)	72 (35-239)	0.81
USAT for PE, n (%)	128.9 (55.9)	139 (59.3)	0.712
Intermediate low-risk, n (%)	109.1 (47.3)	102.7 (43.8)	0.725
Intermediate high-risk, n (%)	88.8 (38.5)	100.4 (42.9)	0.652
High-risk, n (%)	32.5 (14.1)	31.2 (13.3)	0.876

Continuous variables were presented as median and interquartile range (IQR) unless indicated as percentages. *p*-values of <0.05 were considered significant. ^†^ Including all patients with an MDRD-GFR of <60 mL/min. ^††^ Including patients who later developed this complication. ^§^ Signs of right heart systolic dysfunction on computer tomography or echocardiography. ^§§^ Shock on the day of PE diagnosis. Abbreviations: ALT—alanine transaminase; AST—aspartate transaminase; CABG—coronary artery bypass graft; COPD—chronic obstructive pulmonary disease; CVP—central venous pressure; LOS—length of stay; MCS—mechanical circulatory support; NA—not available; PE—pulmonary embolism; USAT—ultrasound-assisted thrombolysis.

**Table 4 jcm-13-02104-t004:** Outcomes after inverse probability weighting.

	Patients with Normal AST/ALT Ratio (n = 230.4)	Patients with Abnormal AST/ALT Ratio (n = 234.34)	*p*-Value
Mortality, n (%) ^†^	6.4 (2.8)	60.1 (25.7)	<0.001 *
Maximum NT-proBNP level (pg/mL)	2400 (982–5067)	3909 (1590–9528)	0.021 *
Shock, n (%)	8.6 (3.7)	88.6 (37.8)	<0.001 *
obstructive, n (%)	1.3 (0.6)	59.5 (25.4)	<0.001 *
septic, n (%)	2.7 (1.2)	4 (1.7)	0.774
haemorrhagic, n (%)	5.6 (2.4)	28 (12)	0.013 *
Acute bleeding-related anaemia, n (%)	49.8 (21.6)	91.7 (39.1)	0.044 *
Acute kidney failure, n (%) ^§^	47.8 (20.8)	94.6 (40.4)	0.027 *
Cardiac arrest, n (%)	21.3 (9.3)	45 (19.2)	0.082
Treatment with inotropes or vasopressors, n (%)	25.7 (11.1)	71.1 (30.3)	0.003 *
Treatment with mechanical ventilation, n (%)	52.6 (22.8)	122.3 (52.2)	0.002 *
Duration of mechanical ventilation (h)	0 (0–0)	4 (0–91)	0.004 *
Treatment with VA-ECMO, n (%)	3.9 (1.7)	22.6 (9.7)	0.006 *
Treatment with CRRT, n (%)	16.7 (7.2)	49.1 (21)	0.048 *

Continuous variables were presented as median and interquartile range (IQR) unless indicated as percentages. * *p*-values of <0.05 were considered significant. ^†^ In-hospital mortality. All patients were discharged to a rehabilitation centre, general ward at another hospital, or home. ^§^ All patients presented with stage 2 or 3 acute kidney failure based on the KDIGO guidelines. Abbreviations: ALT—alanine transaminase; AST—aspartate transaminase; BNP—B-type natriuretic peptide; CRRT—continuous renal replacement therapy; VA-ECMO—veno arterial extracorporeal membrane oxygenation

## Data Availability

Data are unavailable due to privacy and ethical restrictions.

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
