# Peer review of "De Ritis Ratio to Predict Clinical Outcomes of Intermediate- and High-Risk Pulmonary Embolisms"

_jcm, 2024, doi:10.3390/jcm13072104_

Round 1

Reviewer 1 Report

Comments and Suggestions for Authors

The authors investigated impacts of abnormal AST/ALT ratio at baseline on outcomes of (sub)massive PE, and showed that patients with abnormal AST/ALT ratio had worse clinical outcomes even after IPW. I think it is interesting that patients with abnormal AST/ALT ratio were highly likely to develop shock during the clinical course, despite the similar rate of shock at baseline. 

I have a couple of comments. 

I did not understand “obstructive” in tables, which was not mentioned in the text.

In Table 3, age, female sex, and total LOS were duplicated.

Author Response

I did not understand “obstructive” in tables, which was not mentioned in the text.

Answer 1: Dear Reviewer 1, thank you for this important comment. We found that patients had hemodynamic shock due to multiple etiologies and made a differentiation. In this case, it means the classical obstructive/cardiogenic shock from pulmonary embolism. We wrote the subvariables / classification in curved letters to make it understandable for the audience (Table 1 and 3).

In Table 3, age, female sex, and total LOS were duplicated.

Answer 2: Indeed, we duplicated these variables and deleted the double cases (Table 3).

Reviewer 2 Report

Comments and Suggestions for Authors

Interesting correlation between Transaminase levels and clinical outcomes of PE. 

- Suggest to analyze data entirely. In Europe, the term Submassive is not currently used. PE should be stratified into (Classification of pulmonary embolism severity and the risk of early (in-hospital or 30 day) death - Konstantinides SV, Meyer G, Becattini C, Bueno H, Geersing GJ, Harjola VP, Huisman MV, Humbert M, Jennings CS, Jiménez D, Kucher N, Lang IM, Lankeit M, Lorusso R, Mazzolai L, Meneveau N, Áinle FN, Prandoni P, Pruszczyk P, Righini M, Torbicki A, Van Belle E, Zamorano JL; The Task Force for the diagnosis and management of acute pulmonary embolism of the European Society of Cardiology (ESC). 2019 ESC Guidelines for the diagnosis and management of acute pulmonary embolism developed in collaboration with the European Respiratory Society (ERS): The Task Force for the Diagnosis and Management of acute pulmonary embolism of the European Society of Cardiology (ESC). Eur Respir J. 2019 Oct 9;54(3):1901647. doi: 10.1183/13993003.01647-2019. PMID: 31473594.)

Submassive PE as the statement says “ All cases identified automatically were confirm to have a diagnosis of (Sub)massive PE based on computed tomography pulmonary angiogram”. CT pulmonary angiogram the only possible finding to address this diagnosis is regarding RV/LV ratio. Suggest to search data regarding RV dysfunction with ecocardio or patients were included regarding RV/LV ratio in Pulmonary AngioCT.

Regarding the European classification, it is divided into 4 main PE. Low risk (Not included in the manuscript). Intermediate low-risk (Included in the study). Intermediate high risk (Included in the study -Probably Troponins??). High risk or massive PE (included in the study). Suggest to study patients using this classification. RV dysfunction correlates with the elevation of liver enzymes Suprahepatic blood stasis with Inferior vena caval contrast reflux  (Pulmonary AngioCT findings).

High risk or Massive PE (hemodynamic unstable patients) have shown to elevate liver enzymes (Aslan S, Meral M, Akgun M, Acemoglu H, Ucar EY, Gorguner M, Mirici A. Liver dysfunction in patients with acute pulmonary embolism. Hepatol Res. 2007 Mar;37(3):205-13. doi: 10.1111/j.1872-034X.2007.00014.x. PMID: 17362303.

Not only in PE but also in acute Heart Failure (Biegus J, Zymliński R, Sokolski M, Nawrocka S, Siwołowski P, Szachniewicz J, Jankowska EA, Banasiak W, Ponikowski P. Liver function tests in patients with acute heart failure. Pol Arch Med Wewn. 2012;122(10):471-9. doi: 10.20452/pamw.1413. Epub 2012 Oct 4. PMID: 23037318.)

In conclusion if it is possible to address this main concern
Classify patients using Konstantinides et. al classification
analyze data with current parameters risk (PESIs, RV dysfunction, elevated cardiac enzymes).

Comments on the Quality of English Language

No suggestions

Author Response

Interesting correlation between Transaminase levels and clinical outcomes of PE. 

- Suggest to analyze data entirely. In Europe, the term Submassive is not currently used. PE should be stratified into (Classification of pulmonary embolism severity and the risk of early (in-hospital or 30 day) death

Konstantinides SV, Meyer G, Becattini C, Bueno H, Geersing GJ, Harjola VP, Huisman MV, Humbert M, Jennings CS, Jiménez D, Kucher N, Lang IM, Lankeit M, Lorusso R, Mazzolai L, Meneveau N, Áinle FN, Prandoni P, Pruszczyk P, Righini M, Torbicki A, Van Belle E, Zamorano JL; The Task Force for the diagnosis and management of acute pulmonary embolism of the European Society of Cardiology (ESC). 2019 ESC Guidelines for the diagnosis and management of acute pulmonary embolism developed in collaboration with the European Respiratory Society (ERS): The Task Force for the Diagnosis and Management of acute pulmonary embolism of the European Society of Cardiology (ESC). Eur Respir J. 2019 Oct 9;54(3):1901647. doi: 10.1183/13993003.01647-2019. PMID: 31473594.)

Answer 1: Dear Reviewer 2, thank you for your effort and in-depth analysis of our manuscript.

We used the current guidelines of the European Society of Cardiology and Respiratory Society to give more precise information on classification. Therefore, we additionally extracted troponin parameters at admission and identified all patients who had right ventricular pressure and/or volume overload on either echocardiography or computed tomography under the variable “signs of right heart systolic dysfunction”. Furthermore, we scanned the doctoral letters for any signs of hemodynamic instability at hospital admission (Materials and methods, page 2-3 (line 99-105).

We used the new variables to stratify the patients into intermediate low-, intermediate high- and high-risk according to the classification of pulmonary embolism severity and risk of early death in Table 9 of the latest guidelines (Materials and methods, page 3 (line 106-112).

The new variables and the resultant classification are important, perhaps the most important, variables to use in the propensity score calculation. Therefore, we revised our baseline characteristics and confounders to rerun the analysis for inverse probability weighting (Materials and methods, page 2-3 (line 144-145 and Appendix III).

We reported our new results which reflect a more precise comparison and correction for confounders. Particularly, the variables for risk classification were comparable between both groups after IPW (Results, page 7 (line 232), Table 3, Appendices III and IV). 

Submassive PE as the statement says “All cases identified automatically were confirm to have a diagnosis of (Sub)massive PE based on computed tomography pulmonary angiogram”. CT pulmonary angiogram the only possible finding to address this diagnosis is regarding RV/LV ratio. Suggest to search data regarding RV dysfunction with ecocardio or patients were included regarding RV/LV ratio in Pulmonary AngioCT. 

Answer 2: We researched our data and identified all cases with abnormal RV/LV ratio on computed tomography under the variable “signs of right heart systolic dysfunction”. We also used echocardiographic parameters (RVEDD > 44mm, TAPSE < 17mm and ventricular interdependence). Data on echocardiography was available in less patients, therefore, we put all signs of right heart dysfunction under this variable (Materials and methods, page 2-3 (line 99-103), Table 1 and 3).

Regarding the European classification, it is divided into 4 main PE. Low risk (Not included in the manuscript). Intermediate low-risk (Included in the study). Intermediate high risk (Included in the study -Probably Troponins??). High risk or massive PE (included in the study). Suggest to study patients using this classification.

Answer 3: As described in the previous two answers, we addressed this topic and extracted data to make the classification of the current guidelines on pulmonary embolism (Materials and methods, page 2-3 (line 74, 99-112, 115, 144-145)).

RV dysfunction correlates with the elevation of liver enzymes Suprahepatic blood stasis with Inferior vena caval contrast reflux (Pulmonary AngioCT findings). High risk or Massive PE (hemodynamic unstable patients) have shown to elevate liver enzymes (Aslan S, Meral M, Akgun M, Acemoglu H, Ucar EY, Gorguner M, Mirici A. Liver dysfunction in patients with acute pulmonary embolism. Hepatol Res. 2007 Mar;37(3):205-13. doi: 10.1111/j.1872-034X.2007.00014.x. PMID: 17362303. Not only in PE but also in acute Heart Failure (Biegus J, ZymliÅ„ski R, Sokolski M, Nawrocka S, SiwoÅ‚owski P, Szachniewicz J, Jankowska EA, Banasiak W, Ponikowski P. Liver function tests in patients with acute heart failure. Pol Arch Med Wewn. 2012;122(10):471-9. doi: 10.20452/pamw.1413. Epub 2012 Oct 4. PMID: 23037318.) 

Answer 4: Indeed, all kinds of suprahepatic blood stasis from heart failure can increase liver enzymes. However, slight increases in transaminases can be seen frequently without the development of hemodynamic instability or shock. Therefore, the positive predictive value for outcomes was not high in previous studies (Nikolaou et al. 2013, Ambrosy et al. 2013). Our hypothesis is that the De Ritis ratio reflects stronger damage to the liver cells and necrosis which makes it more dependent on time course and burden of diseases.

Therefore, it can offer a rediscovered parameter with a potentially high positive predictive value for mortality and complications. We adjusted and elaborated the pathology of AST/ALT which led to our hypothesis in more detail (Introduction, page 2 (line 57-61)).

To the best of our knowledge, our study investigated the largest cohort regarding this specific hypothesis (Discussion, page 10, line 298-299). 

In conclusion if it is possible to address this main concern

Classify patients using Konstantinides et. al classification 
analyze data with current parameters risk (PESIs, RV dysfunction, elevated cardiac enzymes).

Answer 5: As described previously, we addressed this topic and extracted data to make the classification of the current guidelines on pulmonary embolism (Materials and methods, page 2-3 (line 74, 99-112, 115, 144-145)) which definitely enhanced the quality of our study.

However, we did not have documentation of vital parameters from the day of admission. For this reason, it was not possible to calculate mortality risk based on the original or simplified Pulmonary Embolism Severity Index. Reporting of hemodynamic instability and diagnosis of shock is done retrospectively and there is a risk of selection bias and confounding. We mentioned this limitation in our Discussion (page 11, 383-388).

Reviewer 3 Report

Comments and Suggestions for Authors

The authors wrote a paper evaluating the association of abnormal AST/ALT values with prognosis in pulmonary embolism. The study is very interesting and may have important implications for the management of pulmonary embolism.
My comments:

1. How is (sub)massive PE defined?
2. There is a lack of information on the added value of AST/ALT ratio in the stratification of pulmonary embolism to date: low-risk, intermediate-low risk, intermediate-high risk and high risk of death patients. Perhaps the authors have such data? Then it would be worth adding them.
3. Similarly, it would be valuable to add the assessment of patients on admission on the PESI scale or at least sPESI, and assess the added value of an additional AST/ALT ratio determination.
4. In paragraph 3.2, it should be added that baseline parameters also differed siginificantly in the frequency of chronic heart failure.
5. In the discussion or introduction, I suggest to write more about the difficulties in assessing the prognosis and selection of appropriate treatment in some forms of pulmonary embolism, which was the reason for PERTs (to promote the idea of PERTs among readers).
6. It is worth writing about other interventional methods, such as mechanical aspiration thrombectomy, for which there are currently no clear eligibility criteria, and AST/ALT ratio could serve as an auxiliary indicator - after appropriate evaluation in prospective studies.
7. Limitations should be more clearly established.

Author Response

The authors wrote a paper evaluating the association of abnormal AST/ALT values with prognosis in pulmonary embolism. The study is very interesting and may have important implications for the management of pulmonary embolism. 
My comments:

1. How is (sub)massive PE defined? 

Answer 1: Dear Reviewer 3, thank you for your effort and comments. Before revision, we used this term to describe all patients who were admitted to the hospital. However, this term is currently not used anymore as stated by the previous reviewers and we concluded that all patients were at least at intermediate risk. Therefore, we specified the cohort as patients with intermediate- and high-risk PE to meet the classification of the current guidelines (Title and abstract, page 1 (line 1-3, 19). Introduction, page 2 (line 67). Materials and methods, page 2-3 (line 74)).

  1. There is a lack of information on the added value of AST/ALT ratio in the stratification of pulmonary embolism to date: low-risk, intermediate-low risk, intermediate-high risk and high risk of death patients. Perhaps the authors have such data? Then it would be worth adding them.

Answer 2: As described in the previous answers of reviewer 2, we addressed this topic and extracted data (troponin, parameters of right heart function) to make the classification of the current guidelines on pulmonary embolism as additional variables. Therefore, we rerun the whole statistics and calculated a new pseudopopulation to compare patients with and without abnormal AST/ALT. We wrote a more detailed answer in our first reply to reviewer 2. Changes in the text can be found in Materials and methods, page 2-3 (99-112, 115, 144-145), results, page 5 (line 199-203), results, page 7 (line 232), Table 3, Appendices III and IV.

  1. Similarly, it would be valuable to add the assessment of patients on admission on the PESI scale or at least sPESI, and assess the added value of an additional AST/ALT ratio determination.

Answer 3: Unfortunately, it was not possible to calculate mortality risk based on the original or simplified Pulmonary Embolism Severity Index. We still could extract reporting of hemodynamic instability and diagnosis of shock. This is done retrospectively and there is a risk of selection bias and confounding. We mentioned this limitation in our Discussion (page 11, 383-388).

  1. In paragraph 3.2, it should be added that baseline parameters also differed siginificantly in the frequency of chronic heart failure.

Answer 4: Indeed, this is an important baseline characteristic which differed between both groups. We mentioned it specifically in the Methods, page 5-6, line 204-208. Interestingly, after correction for confounders: chronic left heart failure and all other baseline variables were comparable and did not differ significantly between both groups (Table 3).

  1. In the discussion or introduction, I suggest to write more about the difficulties in assessing the prognosis and selection of appropriate treatment in some forms of pulmonary embolism, which was the reason for PERTs (to promote the idea of PERTs among readers).

Answer 5: We described the challenges and difficultie in assessment for prognosis and treatment in more detail in the Implications section (Discussion, page 10-11, line 345-355). AST and ALT are routinely obtained during hospital admissions. We also believe that the biomarker, after evaluation in RCT, can be used in PERTs to guide treatment decisions.

  1. It is worth writing about other interventional methods, such as mechanical aspiration thrombectomy, for which there are currently no clear eligibility criteria, and AST/ALT ratio could serve as an auxiliary indicator - after appropriate evaluation in prospective studies.

Answer 6: We mentioned the variety of treatment options and (Discussion, page 10-11, line 349-355). We also found that abnormal AST/ALT ratios are predicting higher bleeding risk or even hemorrhagic shock (Table 4). This might be due to multiple pathophysiologic mechanisms (Discussion, page 10, line 330-341). Further studies could evaluate treatment options for patients with higher bleeding risk in this specific group (Conclusion, page 11-12, line 400-404)

  1. Limitations should be more clearly established.

Answer 7: Even though this is one of the only studies on this specific parameter and patient cohort, there are strong limitations which makes it difficult to draw conclusions that can be used in current clinical practice without further investigation. We elaborated limitations for retrospectivity, observer variability and missing parameters in the limitations (Discussion, page 11, line 379-392).

Round 2

Reviewer 2 Report

Comments and Suggestions for Authors

No further suggestions 

Reviewer 3 Report

Comments and Suggestions for Authors

Dear Authors,
thank you for comprehensively addressing my comments and incorporating them into the revisions to the manuscript. I support the publication of the article in its current form.